# Continuity of care, measurement and association with hospital admission and mortality: a registry-based longitudinal cohort study

Øystein Hetlevik ,[1] Tor Helge Holmås,[2] Karin Monstad[2]

[1]Department of Global Public Health and Primary Care, University of Bergen, Bergen, Norway
[2]Research Area Social Sciences, NORCE Norwegian Research Centre AS, Bergen, Hordaland, Norway

**Correspondence to**
Professor Øystein Hetlevik;
Oystein.Hetlevik@uib.no

## ABSTRACT

**Objective** To assess whether continuity of care (COC) with a general practitioner (GP) is associated with mortality and hospital admissions for older patients We argue that the conventional continuity measure may overestimate these associations. To better reflect COC as a GP quality indicator, we present an alternative, service-based measure.

**Design** Registry-based, population-level longitudinal cohort study.

**Setting** Linked data from Norwegian administrative healthcare registries, including 3989 GPs.

**Participants** 757 873 patients aged 60–90 years with ≥2 contacts with a GP during 2016 and 2017.

**Main outcome measure** All-cause emergency hospital admissions, emergency admissions for ambulatory care sensitive conditions, and mortality, in 2018.

**Results** We assessed COC using the conventional usual provider of care index (UPC$^{patient}$) and an alternative/supplementary index (UPC$^{GP list}$) based on the COC for all other patients enlisted with the same preferred GP. For both indices, the mean index score was 0.78. Our model controls for demographic and socioeconomic characteristics, prior healthcare use and municipality-fixed effects. Overall, UPC$^{GP list}$ shows a much weaker association between COC and the outcomes. For both indices, there is a negative relationship between COC and hospital admissions. A 0.2-point increase in the index score would reduce admissions for ambulatory care sensitive conditions by 8.1% (CI 7.1% to 9.1%) versus merely 1.9% (0.2% to 3.5%) according to UPC$^{patient}$ and UPC$^{GP list}$, respectively. Using UPC$^{GP list}$, we find that mortality is no longer associated with COC. There was greater evidence for an association between COC and all-cause admissions among patients with low education.

**Conclusions** A continuity measure based on each patient's contacts with own preferred GP may overestimate the importance of COC as a feature of the GP practice. An alternative, service-based measure of continuity could be suitable as a quality measure in primary healthcare. Facilitating continuity should be considered a health policy measure to reduce inequalities in health.

## INTRODUCTION

Continuity of care (COC) is a declared goal in primary care systems.[1] The concept of COC

---

### STRENGTHS AND LIMITATIONS OF THIS STUDY

⇒ This study is based on detailed nationwide registry data at the patient level for the entire population aged 60–90 years, including all general practitioner (GP) consultations and hospital admissions.

⇒ The analysis presents a novel service-based continuity of care (COC) index, measured at GP practice level, and discusses how this measure relates to the conventional usual provider of care index.

⇒ We analyse how associations vary across gender and levels of education, and how different forms of breaches of COC are associated with patient outcomes.

⇒ A limitation is lack of detailed information of each patient's severity of disease, although we partly compensate for this by including prior healthcare use as a proxy for patients' health status.

⇒ Our data sources do not capture patient experiences with and evaluation of COC.

---

is broad, but most often refers to personal continuity, either measured longitudinally or by standardised questionnaires eliciting information on patient–provider relational continuity.[2–4] Additionally, the continuity concept includes informational continuity and management continuity as measures of level of seamless services across different care providers or sites.[5] In this study, COC reflects the extent to which general practitioner (GP) contacts are concentrated to the same GP in a given time period, which serves as a proxy for relational continuity. Recent literature reviews conclude that continuity is associated with lower mortality[6 7] and a reduction in hospital admissions,[8] indicating better patient treatment in primary care and lower total healthcare costs.[9 10] For chronic conditions considered manageable in primary care (ambulatory care sensitive conditions, ACSCs), hospitalisations are considered a particularly sensitive marker of primary care quality.[11 12] Barker *et al*[13] showed that

longitudinal COC was associated with reduced number of hospital admissions for ACSCs. Despite the acknowledged virtues of COC, there is a steadily increasing policy drive towards giving priority to swift access to care,[14] which may contradict the possibility of seeing a preferred physician and challenge the goal of COC.[13 15]

Since 2001, more than 99% of Norwegian inhabitants have been assigned a regular GP in a list patient system, and as such have a defined usual provider, while also having the option to switch GP twice in a calendar year. Improving continuity in the GP–patient relationship was one of the main arguments for this reform.[16]

In this study, we contribute to the existing literature in several respects. First and foremost, we critically assess the conventional usual provider of care (UPC) index, which we argue may overestimate the association between emergency hospital admissions and COC as a feature of the GP service. The UPC index represents, for each patient, the proportion of contacts with a preferred provider, most often a GP, in relation to all contacts during a given period.[17] Our argument is that the conventional UPC index disregards the fact that when the patient visits another GP than his/her preferred GP, he/she is in many cases in need of acute medical care, as discussed by Lustman et al.[18] Patients in acute need of healthcare are more inclined to be hospitalised; therefore, patient unobserved health is a confounding factor that most likely leads to an overestimation of the (presumably negative) association between COC and hospital emergency admissions. Also, patients may prioritise differently between swift access to any GP and COC with their regular GP, which may influence their relational COC independently of the COC offered at GP service level.[19] We suggest an alternative measure, where COC is disentangled from each individual's unobserved health or personal choices. For each patient, we compute the COC experienced by other patients having the same preferred GP. Our approach is detailed in the methods section.

Second, few studies have investigated how the importance of COC varies by patient characteristics,[7] although COC seems to be of particular importance for patients in vulnerable groups.[20] We measure the association between COC and outcomes for patients with different education levels. Third, we investigate how hospital admission and mortality are associated with different forms of discontinuity of care, where the patients instead of using the preferred GP either use another GP at ordinary opening hours or out-of-hours (OOH) services.

Since 95% of all Norwegian GPs work in group practices,[21] patients may have relational continuity to several GPs who also share electronic health records. This facilitates informational and management continuity when the preferred GP is not available.[22] In contrast, at OOH services, patients meet an unfamiliar physician with no access to the preferred GP's patient records, with low possibility for informational or management continuity to cover up for the broken relational continuity. It is not clear how informational and management continuity is

associated with hospital admissions.[8] Our data allow us to investigate how different kinds of discontinuity of care is associated with hospital admissions as well as mortality. Finally, to our knowledge, this is the first analysis of COC that benefits from patient and GP level data for the entire population.

To sum up, our first aim is to assess whether COC with a GP is associated with mortality and hospital admissions for older patients. We focus on capturing how GP accessibility influences the association and present a GP service-based measure. Further, we aim to explore how the associations between the outcomes and the service-based COC measure vary across gender and educational level. Finally, we aim to analyse how these associations are influenced by breaches of personal COC compared with breaches of personal as well as informational COC.

## METHODS
### Observational period
Several previous studies, for example Barker et al,[13] measure COC and the outcome within the same period. However, following Katz et al,[14] our empirical strategy is measuring continuity over a baseline period of 2 years (in our case, 2016–2017) before a follow-up period of 1 year (2018), when we measure the outcomes.

In this way, we avoid a potential reverse causality problem that arises because hospitalisation may lead to an increase in GP visits shortly after hospital discharge. If these visits are with own GP, COC for the relevant period will be measured as high, and the presumably negative association between COC and hospital admission will be washed out. Vice versa, if postdischarge visits are with another GP, the negative relationship between COC and hospitalisation will be overestimated.

In addition, when applying a baseline period separated from a follow-up period, we acknowledge that greater COC can lead to better disease control and delay or prevent the onset of end-stage disease.

### Data sources
The analysis benefits from administrative registry data at the patient level, covering the entire population, with information on all GP visits in Norway from the database for Control and Reimbursement of Health Care Claims (the KUHR registry). Data originate from reimbursement claims sent by the primary care physician following each consultation, whether at the GP's office or OOH visits at a primary care emergency centre. Since the KUHR registry includes patient identifiers and the consultation date, we can observe, for each patient, the number of primary physician visits during a given period. Furthermore, since the KUHR data also contain GP identifiers, they can be merged with information on each GP's patient list from the Regular GP database. This allows us to identify whether the visit was with the patient's own GP, another GP or at an OOH service.

Data on hospital admissions are provided by the Norwegian Patient Registry and include information on all hospitalisations in Norway: the date of the admission, whether the admission was planned or unplanned, inpatient or outpatient, and diagnoses according to the International Classification of Diseases 10th revision (ICD-10).

From Statistics Norway we obtained data on patient characteristics, including age, sex, education, income, whether the individual lives in a one-person household, and date of death.

## Sample selection

Since COC is regarded especially important for patients with chronic conditions, which are more widespread in the elderly population, we restrict the sample to the age group 60–90. The number of inhabitants in this age group by 1 January 2016 is 1 100 010, of which nearly all, 1 099 268, are registered with a regular GP, while the group of regular GPs at this point counts 4644. We apply the following sample selection criteria:

► The sample is restricted to patients who belong to the same GP–patient list in the baseline period. This criterion ensures that a unique preferred GP can be defined (in total 947 153 patients) and implies that we only include GPs practising by 1 January 2016 and throughout the baseline period (3990 GPs).
► We exclude patients who die within the baseline period (53 366 patients).
► We restrict the sample to patients who have at least two consultations with primary care physicians during the baseline period (in total 757 873 patients and 3989 GPs). This is in line with most previous studies using UPC as a measure of continuity.[2 13]

## Outcomes

Our dependent variables are three-fold, and all outcomes are measured in 2018: (1) the number of *all-cause acute hospital admissions,* (2) the number of *acute hospital admissions for ACSCs,* where the definition of ACSC is adapted from Bardsley et al,[23] also applied in Barker et al.,[13] (3) *mortality,* a variable that equals 1 if the individual dies during 2018.

## COC measures

We apply two alternative definitions of the provider of care index, labelled $UPC^{patient}$ and $UPC^{GP\ list}$. Notably, in accordance with Norwegian patient policy, we define 'the usual provider' as the regular GP that the patient is enlisted with. The UPC$^{patient}$ index is the traditional index applied in for instance Barker et al.[13] For instance, if an individual has 10 contacts in total, and 6 of them are with his own GP, the UPC$^{patient}$ score will be 0.6.

In order to disentangle the COC measure from individual *i*'s unobserved health and background characteristics (as explained in the introduction), we construct continuity measure UPC$^{GP\ list}$. This index is based on the COC experienced by *patients other than i* belonging to the same GP list population. That implies that for each

patient, we use information on the continuity experienced by other patients enlisted with *i*'s GP. An example could be a group of three patients A, B, C enlisted with the same GP. If A has 8 out of 10 visits with own GP, B has 3 out of 7, and C has 12 out of 12, then A's UPC$^{GP\ list}$ index score would be (3+12)/(7+12), B's score would be (8+12)/(10+12) and C's score would be (8+3)/(10+7). The UPC indices are in the interval (0,1), with a higher score reflecting greater COC.

Formally, the $UPC^{GP\ list}$ index can be expressed as follows:

$$UPC_i^{GP\ list} = \frac{X_{j*}^{-i}}{\sum_{j=1}^{J} X_j^{-i}}$$

where *i* is an individual, *j* is one of *J* GPs, *j\** is the GP whom individual *i* is enlisted with and $X_j^{-i}$ is the number of visits to GP *j* of all individuals enlisted with j*, except for *i*. Using this notation, the $UPC^{patient}$ index is defined as

$$UPC_i^{patient} = \frac{X_{j*}^i}{\sum_{j=1}^{J} X_j^i}$$

The $UPC^{GP\ list}$ index is a general concept, applicable irrespective of the size of the GP list population. If GP list populations are large, the $UPC^{GP\ list}$ index will be very similar to an index at GP level where all patients are included. We have estimated using such a simpler index, and results are virtually identical (not reported here).

When data availability allows it, the COC analysis can include an investigation of the impact of different breaches of personal COC. For this purpose, we construct two indices of *dis*continuity: the proportion of all consultations in a GP list population that is with another GP at ordinary opening hours (Other–GP index) and the proportion of OOH visits (OOH–index), respectively.

## Covariates

To control for patient heterogeneity, we include information on a number of individual demographic and socioeconomic characteristics: sex, age, level of education in three categories (primary, secondary and higher education), gross total income in Norwegian Crowns (NOK) 100 000, and an indicator for one-person household (measured in 2016). Furthermore, utilisation of healthcare in the baseline period (2016–2017) may be indicative of the individual's health status. As controls, we therefore include the number of GP visits, the number of all-cause hospital admissions (whether planned or unplanned), the proportion of inpatient admissions, as well as the proportion unplanned hospital admissions.

## Statistical analysis

We used multivariable regression with municipality-fixed effects (the municipality where the GP practice is located), to test the association between COC and the number of hospital admissions and mortality. To reduce bias due to population heterogeneity, we included a series of covariates that may influence the

association: individual demographic and socioeconomic characteristics as well as health status indicators, as detailed above. Furthermore, a fixed effects approach can reduce the impact of unmeasured confounding. Since the fixed effects estimator relies only on variation within the chosen unit, this estimator is not affected by confounding from unmeasured time-invariant factors. Notice that the term 'fixed effects' is distinct from that of the statistical literature, where this term typically is used in the context of random effects or mixed models describing a parameter associated with an entire population (see eg, Gunasekara *et al* for a non-technical presentation of the fixed effects estimator and a comparison with mixed models).[24] In the context of this analysis, the (municipality-) fixed effect estimator controls for any time-invariant differences between municipalities, such as centrality, which may influence the availability of GPs—in particular, experienced GPs—as well as distance to hospital and other specialist health services. Since our outcomes are either count data (number of admissions) or binary (mortality), non-linear models such as count data models or logit models may seem appropriate. However, estimating a fixed-effect logit model resulted in numerical overflow (due to too many effective observations). Therefore, we have chosen to estimate linear fixed effect models rather than non-linear models without fixed effect.

In stratified analyses, we estimate the associations separately by gender and by level of education. In an additional estimation, we use the two indices of *dis*continuity as main explanatory variables, while otherwise, the model specification is kept unchanged.

The ordinal covariates, such as age and number of GP visits, are included in their natural units, apart from income, which enters in NOK 100 000s. As a sensitivity analysis, we have estimated linear splines models, which gave virtually identical results (for details, see online supplemental file A). The COC index enters the model linearly, which is common in the literature.

The relationship between COC and our outcomes is complex, and we do not claim that the current analysis detects causal pathways. For instance, it is conceivable that GPs offering greater COC attract a higher proportion of patients with chronic conditions, which would imply a selection bias. Still, we have tried to address the identification issues in several ways. First, we present an alternative index that we argue modifies the impact of patient unobserved health on observed COC. Second, information on a rich set of covariates helps reflect the individual's health status. Third, we observe these control variables in a baseline period *pre* outcomes. Finally, to improve inference, we take into account that patients enlisted with the same GP most likely share some (unobservable) characteristics that may influence outcomes, such as health,

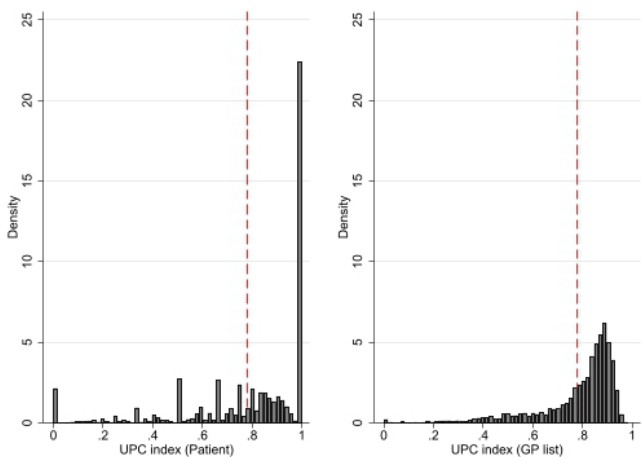

**Figure 1** Distribution of usual provider of care indices (n=757 873) based on data from individual patients (left figure: (UPC^patient)) and at general practitioner (GP) practice level (right figure: UPC^GP list). Red dotted lines: mean value of each index. UPC, usual provider of care.

social background, and a preference for a certain GP practice style. These characteristics may not be fully adjusted for by municipality fixed effects; therefore, we cluster standard errors at the GP level.

Analyses were performed using STATA V.16.

### Patient and public involvement
Patients or public were not involved in the design or interpretation of this study.

## RESULTS
In total, 757 873 individuals, enlisted with 3989 GPs, met our inclusion and exclusion criteria. In the following, we will compare results from using the conventional UPC index to results from using the service-based COC measure presented above.

Figure 1 reveals that for these two indices, mean scores are similar, yet there are large differences in the distribution. Note that for 22.5% of the sample, the UPC^patient score equals one, that is, for these patients, all their primary physician visits during the baseline period were with their own GP. Also, about 2.5% of the sample had no visits to own GP, that is, a UPC^patient score of zero. By construction, the values zero and one do not appear when we apply the UPC^GP list index.

Table 1 reports summary statistics of outcome variables and covariates. We see that 53% of the sample are women, the mean age is 70.7 years and 35% live alone. The overall use of healthcare is high, about 9.4 GP visits and 5.1 hospital admissions during the 2 years' baseline period. This is to be expected, given the high age and the fact that a certain (low) level of GP use is an inclusion criterion. In the baseline period, the mean number of emergency hospital admissions per year is about 0.3 (11.9% of 5.1 hospital contacts were emergency cases during these 2 years), while the mean number of all-cause

**Table 1** Descriptive statistics, patients aged 60–90 with at least two general practitioner (GP) visits in 2016–2017

| | |
|---|---|
| Continuity of care (measured at baseline, 2016–2017): | |
| UPC$^{patient}$ * | 0.788 (0.263) |
| UPC$^{GP list}$ * | 0.780 (0.169) |
| Patient characteristics (measured at baseline, 2016–2017): | |
| Proportion females | 0.531 |
| Age | 70.675 (7.562) |
| Proportion low education† | 0.259 |
| Proportion medium education† | 0.498 |
| Proportion high education† | 0.243 |
| Total income (100 000 NOK) | 4.310 (5.067) |
| Proportion one-person household† | 0.353 |
| No. of GP visits | 9.432 (7.967) |
| No. of hospital in- and outpatient contacts | 5.103 (9.884) |
| Proportion inpatient stays | 0.093 |
| Proportion emergency admissions | 0.119 |
| Number of patients | 757 873 |

Proportions or means with standard deviation in parentheses.
*UPC$^{patient}$ and UPC$^{GP list}$: usual provider care index measured for each patient and for each GP practice, respectively.
†Measured in 2016.
ACSC, ambulatory care sensitive conditions; UPC, usual provider of care.

hospital admissions in the follow-up year is slightly higher, 0.357. For the two COC measures, we see that the mean values are very similar (0.78), but the SD is larger for the UPC$^{patient}$ index, as can also be seen from figure 1.

Irrespective of COC measure (table 2), we find a negative relationship between COC and emergency admissions, and notably, the estimated association is far

stronger with the UPC$^{patient}$ than with the UPC$^{GP list}$. Using the latter index, no relationship between continuity and mortality emerges. The following illustrates the difference in magnitude of estimates: when the index score increases by 0.2, the number of emergency admissions for ACSC is reduced by 8.1% when measured with the UPC$^{patient}$ and by merely 1.9% when measured with UPC$^{GP list}$. As expected, the relationship between COC and emergency admissions is stronger for ACSC than for other conditions, irrespective of index used.

The coefficient estimates of the control variables show the same pattern across outcomes: the number of emergency admissions as well as mortality is higher for males than for females, increasing in age, decreasing in education, virtually independent of income, and higher for individuals living in one-person households than for married/cohabitating. Individuals who are frequent users of GP or hospital visits in the baseline period have a higher number of hospital admissions and a higher mortality risk the following year.

### Heterogeneity

In the following, we will investigate whether the relationship between COC (measured by the UPC$^{GP list}$ index) and our outcomes depends on patient characteristics. When we estimate separately by gender, we get qualitatively the same results as reported in table 2, although with lower levels of significance. There is no clear difference in results between men and women (we have included an interaction term between the COC measure and gender, and its coefficient is not statistically significant, not reported here). Next, we estimate separately by the three levels of education. Regarding mortality, we find no statistically significant association for any of the subsamples defined by educational level. We therefore focus on the relationship between COC and hospital admission.

From table 3, we learn that the association between COC and all-cause emergency admissions is far stronger

**Table 2** Associations between continuity of care and the outcomes emergency admissions to hospital and mortality

| | Emergency admission | | Emergency admission for ACSC | | Mortality | |
|---|---|---|---|---|---|---|
| | UPC$^{patient}$ index† | UPC$^{GP list}$ index† | UPC$^{patient}$ index† | UPC$^{GP list}$ index† | UPC$^{patient}$ index† | UPC$^{GP list}$ index† |
| Index score | −0.104*** | −0.028** | −0.027*** | −0.006* | −0.010*** | 0.000 |
| | (−22.16) | (−3.78) | (−15.67) | (−2.51) | (−13.42) | (0.28) |
| $R^2$ (within) | 0.088 | 0.088 | 0.049 | 0.049 | 0.046 | 0.043 |
| Mean Y | 0.357 | 0.357 | 0.068 | 0.068 | 0.024 | 0.024 |
| Relative effect‡ | −5.8% | −1.6% | −8.1% | −1.9% | −8.3% | 0.2% |
| N | 757 873 | 757 873 | 757 873 | 757 873 | 757 873 | 757 873 |

In all estimations, control variables include individual patient characteristics listed in table 1 and municipality-fixed effects.
$t$ statistics in parentheses. *p < 0.05, **p < 0.01, ***p < 0.001.
†UPC$^{patient}$ and UPC$^{GP list}$: Usual provider care index measured for each patient and for each GP practice, respectively.
‡Percentage change in outcome when the index increases by 0.2.
Estimated with two different indices, standard errors clustered at GP level.
ACSC, ambulatory care sensitive conditions; GP, general practitioner; UPC, usual provider of care.

**Table 3** Associations between continuity of care measured at GP level (UPC $^{GP\text{-}list}$) and emergency admissions to hospital, estimated for each patient in the list population, stratified analyses by educational level

| | Emergency admission | | | Emergency admission for ACSC | | |
|---|---|---|---|---|---|---|
| | Primary education | Secondary education | Higher education | Primary education | Secondary education | Higher education |
| UPC $^{GP\text{-}list}$ Index score | −0.065*** | −0.028** | 0.010 | −0.009 | −0.007* | −0.000 |
| | (−4.42) | (−2.73) | (0.78) | (−1.43) | (−2.09) | (−0.18) |
| $R^2$ *(within)* | 0.093 | 0.086 | 0.075 | 0.011 | 0.046 | 0.038 |
| Mean Y | 0.440 | 0.342 | 0.298 | 0.099 | 0.063 | 0.043 |
| Relative effect† | −2.6% | −1.3% | 0.5% | −1.6% | −1.9% | −0.3% |
| N | 196 482 | 377 635 | 183 756 | 196 482 | 377 635 | 183 756 |

t statistics in parentheses. *p < 0.05, **p < 0.01, *** p < 0.001.
Control variables include individual patient characteristics listed in table 1 and municipality-fixed effects.
†Percentage change in outcome of a one SD change in the index score.
Estimated with the UPC$^{GP\ list}$ index and standard errors clustered at GP level.
ACSC, ambulatory care sensitive conditions; UPC, usual provider of care.

for patients with primary education than for the two other categories. There is no such socioeconomic gradient for the outcome emergency admissions for ACSC.

### Inspecting different forms of discontinuity of care

So far, our explanatory variable of interest has been the continuity with which patients in a GP list population visit their own GP, while we have not distinguished between visits to other providers of primary doctor services. We now change the focus of the analysis, to investigate the relationship between the outcomes studied and different forms of *dis*continuity of care. In our data, we can identify two categories of such consultations: visits to another GP at ordinary opening hours (a partner, registrar or locum), and OOH visits. As reported in table 1, about 79% of all consultations are with own GP, while about 17% are with another GP, and 5% are OOH visits. We also find that GPs rarely treats patients belonging to a list outside their group practice since such cases qualify for a particular fee, triggered in merely 0.6% of consultations.

In the following, we will compare outcomes for individuals belonging to list populations that, to different degrees, visit other GPs or OOH services, respectively. We find this interesting to investigate, since different forms of broken COC bear different implications for informational and management continuity and probably also relational continuity. In this part of the analysis, we therefore change our specification by replacing our COC measure with two new explanatory variables: (1) other-GP index (the proportion of GP visits that is with another GP than the preferred GP during ordinary opening hours), and (2) the OOH-index (the proportion of GP visits that is OOH visits). Both proportions are out of the total number of GP and OOH doctor visits in the GP list population.

As before, we estimate outcomes at the individual level, controlling for municipality-fixed effects and individual

patient characteristics related to the baseline period. Results are reported in table 4.

We find that—for an individual patient in the list population—belonging to a list where OOH visits are more frequent is associated with a higher risk of both kinds of emergency admissions. On the other hand, emergency admissions for ASCS and mortality are both unrelated to the frequency (proportion) of consultations done by other GPs within their ordinary practice.

### DISCUSSION
### Main findings

In this nationwide registry-based study using individual-level data from Norwegian healthcare, we investigated COC as predictor of acute hospital admissions and mortality among patients aged 60–90 years. We compared two measures of COC: the conventional UPC index (UPC$^{patient}$) and an index defined for each GP list population (UPC$^{GP\ list}$). In line with earlier studies, we find a marked association between UPC$^{patient}$ and lower rates for acute admissions and mortality. Both indices were inversely associated with all-cause emergency hospital admissions and emergency admissions for ACSC. However, the associations between COC and these outcomes were considerably weaker when COC was measured with the index defined at GP level. We found no association between mortality and the UPC$^{GP\ list}$ index, while there was an estimated 8.3% decrease in mortality with a reduction of 0.2 in the conventional UPC index.

Analysing subsamples defined by level of education, we found a clear gradient with the strongest association between COC and all-cause hospital admissions for patients with low education. No such gradient was found for ACSC admissions.

Other things equal, emergency admissions are more common for individuals who belong to a GP list

**Table 4** Associations between proportion of consultations with other general practitioners (GPs) or out-of-hours (OOH) services in list population of the patient's regular GP and emergency admissions to hospital and mortality estimated at individual patient level

| | Emergency admission | Emergency admission for ACSC | Mortality |
|---|---|---|---|
| Other-GP index | 0.014* (1.99) | 0.002 (0.71) | −0.000 (−0.47) |
| OOH-index | 0.959*** (9.57) | 0.285*** (7.45) | 0.017 (1.49) |
| $R^2$ *(within)* | 0.088 | 0.048 | 0.046 |
| Mean Y | 0.357 | 0.068 | 0.024 |
| Relative effect variable 'Other GP index'† | 0.6% | 0.5% | −0.2% |
| Relative effect variable 'OOH-index'† | 5.3% | 8.3% | 1.4% |
| N | 757 873 | 757 873 | 757 873 |

t statistics in parentheses. *p < 0.05, **p < 0.01, ***p < 0.001. Control variables include individual patient characteristics listed in table 1 and municipality-fixed effects. Standard errors clustered at GP level.
†Percentage change in outcome of a one SD change in the index score.
ACSC, ambulatory care sensitive conditions.

population where the use of OOH visits is frequent. We found no or minimal increase in number of emergency admissions if patients, instead of using their preferred GP, visited another GP. Data tell us that the 'other GP' very rarely treats patients from outside the patient lists of his/her group practice. Therefore, we infer that, as a rule, the other GP and the preferred GP work within the same GP practice.

### Interpretation of results

There is a policy drive towards reducing acute hospital admissions to reduce healthcare costs and streamline health services, and a well-functioning primary healthcare is judged to be important to achieve this.[25] Earlier studies have found that COC is associated with a reduction in all-cause acute hospital admissions[10 14] as well as acute admissions for ACSCs,[13] and our results support these findings. Recent studies report an inverse association between COC and mortality.[7 26] In our study, we found this result only when using the conventional UPC index. The lack of (a negative) association between COC at practice level and mortality partly contradicts earlier findings.[27 28]

The literature assessing the impact of COC on secondary healthcare utilisation mainly applies data on the individual patient and his/her use of the preferred GP. Our study supports the positive association between such measures of COC and positive health outcomes. However, in this paper, we add another perspective, by introducing an alternative index which reflects the COC in the preferred GP's practice population. Using this UPC[GP list] index, we found that the strength of COC as a predictor for admission attenuated markedly, and the UPC[GP list] index was not associated with mortality. These differences in associations between COC and the outcomes indicate that some patient characteristics or choices are reflected in the conventional UPC index that add to the effect of COC as a measure of healthcare performance. Even though we adjust for prior care contacts in our analyses, variation in

patient unobserved health may influence the UPC[patient] index. A patient with more unstable disease is more often in need of swift access to care and might therefore more often visit other GPs or OOH services, and more often need a hospital admission, as also discussed by Lustman *et al*.[18] The result is a lower conventional UPC index and a higher frequency of admissions, thereby increasing the COC effects, partly independent of the COC offered by the preferred GP during ordinary opening times. In addition, patients differ in their ability to manoeuvre in the healthcare system. Resourceful patients, who may also deal better with their diseases, may achieve a high COC even if the system level COC is low,[19] and thereby also achieve the benefit of COC. This possibility is of course related to how the service is organised but will to a large extent reflect the patient's personal choices. A UPC index based on the individual patient's contacts with healthcare might overestimate the effect of COC as a feature of the GP practice, and a service-based measure of COC might be a more useful marker of care quality.

Our finding that the strength of the association between COC and all-cause acute hospital admissions is inversely related to educational level possibly reflects socioeconomic differences in ways of using healthcare. An explanation could be that patients with lower education tend to be more dependent on the GP facilitating COC to help them achieve the benefit of better continuity,[20] in line with the theory of advocacy set forward by Donaldson.[29] Also, multimorbidity is more common among patients with low education, and based on previous studies, COC is of greater importance under such conditions.[30] However, we found no educational gradient with respect to acute ACSC hospital admissions, which could indicate a more equally distributed GP service related to ACSC, with standard guidelines and scheduled follow-up programmes.

The present study shows that the number of emergency hospital admissions is almost unaffected by other GPs

taking a larger share of consultations in the list population. On the other hand, there was a marked increase in frequency of hospital admissions if the UPC$^{GP list}$ was lower due to more frequent use of OOH services by patients in the list population. This indicates that the benefits of COC can be reaped also when using other GPs in a group practice, either because patients are familiar with more GPs ensuring relational continuity, or the lack of personal continuity can be compensated for by informational continuity.[22] This finding supports a policy for increasing accessibility to GPs in daytime practices where a smaller group of GPs share responsibilities.

### Strength and limitations

This analysis of COC is based on patient level data for the entire population, including all GP consultations and hospital admissions, combined with comprehensive data on patient characteristics and prior use of healthcare. In this analysis using claims data, health-related non-response is not a valid concern. Our findings are generalisable internationally in contexts where primary care is the first point of contact.

The longitudinal approach is also a strength, with COC estimation based on a 2-year period prior to the 1-year period when outcomes are assessed. A limitation is lack of information of each patient's severity of disease, although we partly compensate for this by including prior healthcare use as a proxy for health problems. A major weakness with registry data is that patient experiences with and evaluation of COC are not captured, whether at individual level or at GP list population level. In the discussion about the connection between COC and hospital admissions, an underlying assumption is that reduced admission rates is a positive outcome, especially for ACSC where there is a special potential to stabilise patients with good routine care, and hospital admission rates are considered a performance indicator of primary healthcare. Still, based on data in the present study, we cannot know if the reduced frequency of hospital admissions benefits the individual patient, except we observe no increase in mortality.

In this analysis, we compare the conventional COC index to an alternative index. There are several indices in use, which are highly correlated.[2 13] We chose the conventional UPC index as a starting point precisely because it is often used, the most straightforward to interpret and possible to adapt to estimation of COC delivered in a healthcare organisation, in this case a GP practice. A COC measure that is more directly measuring performance of care, such as the proposed UPC$^{GPlist}$ index, has a potential to be a useful quality measure, we argue.

### Implications and further research

The UPC$^{GP list}$ index is significantly associated with acute hospital admissions and could be a possible quality measure to inform policy decisions and follow-up policy changes if the goal is to ensure COC in primary care. COC at practice level could be estimated using national registry data as in the present paper or by data available

within a practice, as Sidaway-Lee *et al* describe.[31] However, whether a maximal gain of COC is achieved, seems to depend on patient characteristics in addition to healthcare performance. This is an important finding in relation to current changes in several healthcare systems, which emphasise quick access, private service providers and private insurance-based services for groups that may afford to buy these services, at risk of losing benefit of COC and probably increased total healthcare costs. Altogether, it seems that the healthcare system needs a primary care organisation that facilitates COC but also patients who are informed about the benefit of COC and willing to use services so they might achieve COC, but there is need for further research on how to promote COC. COC seems to be especially important for patients with low education, and prioritisation of COC by GPs and healthcare policy might be a contribution to reducing inequalities in health. A main goal for the list patient system is to provide equitable healthcare for all inhabitants, and variation in COC and the related consequences shown is unwarranted variation that should be reduced.

However, since our suggested index represents a new approach, it needs further assessment in research. It would be interesting to see if studies based on registry data from other contexts give similar results. Also, qualitative studies can give more in-depth understanding of various aspects of the COC concept and how patients and professionals view different COC measures. If survey information can be merged with registry data on healthcare utilisation and GP lists, it should be possible to compare patients' reports with the two COC measures.

GPs' personal responsibilities are extensive and work pressure is high. The expectation to deliver continuous care with high accessibility is increasingly problematic when recruiting and maintaining GPs.[32 33] Our findings indicate that GPs organised in group practices might achieve the same benefits of COC for their patients as a longitudinal, personal continuity with one regular GP, at least regarding the number of acute hospital admissions. This finding might support a stronger focus on common responsibilities between GPs in the future, but changes should be followed by more research on effects.

**Contributors** THH and KM outlined the study in the application for the funding grant. Together with ØH they further developed the project idea. THH had the main responsibility for analysing data, and all authors contributed to the interpretation of data. ØH and KM drafted the article, all authors reviewed and edited the manuscript, and approved the version that is submitted. ØH is the guarantor of the study.

**Funding** This study is part of a project funded by The Research Council of Norway, project number 288592.

**Competing interests** None declared.

**Patient consent for publication** Not applicable.

**Ethics approval** The study is approved by Committee for Medical and Health Research Ethics (Project number 2019/771/REKvest). The ethics committee waived the need for patient consent for the use of these registry data. Data were linked by Statistics Norway and pseudo-anonymised before delivered to the researchers.

**Provenance and peer review** Not commissioned; externally peer reviewed.

**Data availability statement** Data may be obtained from a third party and are not publicly available. Data are available from the national registries on a reasonable

request. The use of data in this study is approved by application for this specific project only and data cannot be shared with anyone outside this project.

**ORCID iD**
Øystein Hetlevik http://orcid.org/0000-0001-8912-3426

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
