## [Reviewer comments · BMJ Open]

ARTICLE DETAILS

TITLE (PROVISIONAL)	Continuity of care, measurement and association with hospital admission and mortality – a registry-based longitudinal cohort study
AUTHORS	Hetlevik, Øystein; Holmås, Tor; Monstad, Karin

VERSION 1 – REVIEW

REVIEWER	Baker, Richard University of Leicester, Health Sciences
REVIEW RETURNED	08-May-2021

GENERAL COMMENTS	The points raised in my review of this paper for the BMJ have been largely addressed. The English is generally fine, but some minor improvements in the interests of clarity could be made during editing. The main weakness of the study remains the use of a measure of continuity that has not been evaluated and it is therefore difficult to know how much weight to place on the findings using the UPCList measure. The authors acknowledge that further research is needed before the properties of this measure are known. I agree, but suggest that some text might be added to explain what research is needed. UPCpatient is arguably the gold standard for measuring the concentration of care in a single provider but there are other measures against which UPCList could be compared, including patient reports of continuity which provide a patient view on relational continuity. Face validity could be explored by seeking the views of patients and professionals on the meaning of UPCList. Qualitative studies have potential to explore some of the differences between the measures used in the study. The statement 'needs further research assessment' is rather brief as a signal to what needs doing next. A reservation about UPCList is that it describes the continuity experienced by the patients of one doctor, excluding the patient of interest. It's a measure of the service rather than the care the patient has received. The authors may like to consider whether use of a ratio (UPCpatient : UPCList) would provide an alternative measure that is more readily interpreted. Patients whose experience is different to the norm may have different outcomes. It can be hypothesised that patients who experience less continuity than the other patients of the same doctor may be at increased risk of admission or mortality; it's possible than higher continuity than other patients would be associated with more consistent care and lower risk of admissions or death or that this high continuity group include people towards the end of life who are at greater
---

	risk of admission or death. To be clear, I am not suggesting that the authors run this analysis for this paper (I should have thought of this when first reviewing the paper for the BMJ) but it may be something they would want to think about.
--	---

REVIEWER	Skjøth, Flemming Aalborg University Hospital
REVIEW RETURNED	25-Jun-2021

GENERAL COMMENTS	The authors Dr. Hetlevik et al present a study on measure for continuity of care based on Norwegian nationwide register data. The authors argue that continuity of care should(/could) be regarded as a feature of the GP-practice to which the individual patient/citizen is related, and it is argued that traditionally continuity of care is viewed as a patient level feature. From an epidemiological point both levels may be relevant, but importantly they reflect different questions. If the focus is on patient administration and service on GP level, then indeed a GP-level index seems appropriate. It is not entirely clear from the text how GPs and practices are treated, the text could be strengthened by being more consistent on use of practice and GP list. Perhaps you could provide some additional information on the number of practices and fix the proportion of 1-GP practices. Which may be relevant for the evaluation of the change of not being assigned to the regular GP or practice. The suggested index UPC^{GPlist} is the average number of visits at the regular GP among total number of visits excluded the visits of the current patient, as opposed to the index $UPC^{patient}$ being based only on the patients own visits. The principle of excluding the patients own data in the index is well-thought, although for a practice/GP with moderate amount of patients (average approx. 220) then the index would probably not vary much between patients within practice/GP. Hence a sensitivity analyses could suggested on a simpler index being just the GP level average number of visits at regular GP. The index is calculated over a two-year period 2016-17 and assumed indicative for the service in 2018 in which the individual level outcomes are observed. This seems appropriate for this purpose. The comment p9 l7 must relate to both indexes. In the description of included covariates a number of non-categorical covariates are listed, it should for transparency of the analysis be specified how these are included. As they are potential confounders their effects are of no interest in respect to the manuscript, a flexible approach could then be applied such as splines for non-categorical covariates. It is not clear what the purpose of the comment regarding municipality-fixed effects is, are the included and if so what are they. Furthermore as the analyses are clustered at practice/GP level then these effects are to some extend already accounted for.
--

	It also need detailed how the exposure index UPC actually enters the model, I presume a linear relation is assumed. Was this assumption evaluated? As the UPC^GP index is a GP level index, I think the table 1 presentation could be strengthened by giving more information on GP level as eg number of practices/GP, number of patients per practices/GP, number of GPs per practice. Listing the outcomes in table 1 is in-appropriate in my view, you also repeat them in table 2. As regards table 2 all listing of the effects of patients characteristics should be taken out, they are of no interest for your investigation of the effects of the continuity of care index, and in general you have no guarantee that the study design and other included covariates would ensure that the parameters have the anticipated interpretation. The analysis represented by results in table 3 should be described in the methods section, I assume they are obtained by stratified analyses. A final explorative analysis is presented, it is not clear what the purpose of these are and they seem to have another purpose than these primary exposure index. Perhaps it should be clarified whether the indexes analyzed here are patient characteristics or practice/GP characteristics. Minor What is ACSC? It disturbs that the tables 2-4 although being structured likewise then the rows are permuted in table 4.
--	---

REVIEWER	Faisal, Muhammad University of Bradford
REVIEW RETURNED	29-Jun-2021

GENERAL COMMENTS	Hetlevik and Holmås aimed to explore the association of continuity of care with hospital admission and mortality. Overall study has used appropriate statistical methods, but the results can be presented in better form, for example, distribution of covariates and their relationship with outcomes can be reported. Following are my detailed comments:  - I thought a research question was to explore the association of two COC indices (conventional vs proposed) with outcomes such as mortality and hospital admission. However, suddenly exposure is changed to 'Education level'. A well-defined research question is needed. - I appreciate this study just explore association, which is different from a causal relationship, but interpreting all model coefficient will produce biased results and it is well known as Table 2 fallacy Daniel Westreich, Sander Greenland, The Table 2 Fallacy: Presenting and Interpreting Confounder and Modifier Coefficients, American Journal of Epidemiology, Volume 177, Issue 4, 15 February 2013, Pages 292–298, https://doi.org/10.1093/aje/kws412 - Ideally, table 1 shall split up by outcome. - How the relationship of outcome with covariates has been explored? Boxplot or scatter plots?
--

VERSION 1 – AUTHOR RESPONSE

Reviewer: 1 Prof. Richard Baker, University of Leicester	
Comments to the Author: The points raised in my review of this paper for the BMJ have been largely addressed.	
1.1 The English is generally fine, but some minor improvements in the interests of clarity could be made during editing.	Our text has been language-checked professionally and we have reread the text thoroughly and hope it is now up to standards.
1.2 The main weakness of the study remains the use of a measure of continuity that has not been evaluated and it is therefore difficult to know how much weight to place on the findings using the UPCList measure.	We agree that the UPC GP list index is not evaluated before and expect and appreciate scepticism about the usefulness of this as a measure of continuity. The GP list index is presented in a transparent way so it can be evaluated in other studies, which we of course hope will happen. It is also our understanding that the conventional UPC has become a standard not through an evaluation or validation process, but because it has proven to be associated with different wanted outcomes. This UPC at patient level is often presented as a measure quality of performance of care in general, but the precision of UPC interpreted in this way may be questioned since UPC clearly is influenced by the patient’s preferences, choices and health. COC is a very “hot topic” in health care development in countries with strong primary care, and COC is challenged because of increasing complexity in care. Therefore, in our opinion, it is valuable to try to find different ways to evaluate care. This motivated us to look at COC with a focus on the service provider, in this study the GP, and study associations with desired outcomes of care. All this said, we consider UPC-patient a useful measure of COC and have made some changes in manuscript to underline the clear associations we find between UPC-patient and reduces mortality and admissions. See page 18 &20
1.3 The authors acknowledge that further research is needed before the properties of this measure are known. I agree, but suggest that some text might be added to explain what research is needed. UPCpatient is arguably the gold standard for measuring the concentration of care in	We appreciate these concrete suggestions. In the subsection “implications and further research ” (page 23-24) in the Discussion section, we have elaborated on the issue of further research, inspired by your comments. The lack of patient reports is also listed among the 5 bullet points regarding strengths and

a single provider but there are other measures against which UPCList could be compared, including patient reports of continuity which provide a patient view on relational continuity. Face validity could be explored by seeking the views of patients and professionals on the meaning of UPCList. Qualitative studies have potential to explore some of the differences between the measures used in the study. The statement 'needs further research assessment' is rather brief as a signal to what needs doing next.	limitations.
1.4 A reservation about UPCList is that it describes the continuity experienced by the patients of one doctor, excluding the patient of interest. It's a measure of the service rather than the care the patient has received.	We agree, but that is also the intention. This a method chosen to reduce the impact of patient "personal factors" that could be a weakness of UPC-patient as a measure of care performance. And therefore, we argue that UPC-GPlist probably is a better measure of the service. It is an assumption in this approach that patients in question/individual patient is given "services as usual" by their GP, i.e COC as offered to the list population
1.5 The authors may like to consider whether use of a ratio (UPC_{patient} : UPCList) would provide an alternative measure that is more readily interpreted. Patients whose experience is different to the norm may have different outcomes. It can be hypothesised that patients who experience less continuity than the other patients of the same doctor may be at increased risk of admission or mortality; it's possible than higher continuity than other patients would be associated with more consistent care and lower risk of admissions or death or that this high continuity group include people towards the end of life who are at greater risk of admission or death. To be clear, I am not suggesting that the authors run this analysis for this paper (I should have thought of this when first reviewing the paper for the BMJ) but it may be something they would want to think about.	We thank the reviewer for this very interesting proposal. Such an approach might further clarify the relationship between COC as a characteristic of the GP service and COC related to personal choices or ability to find their way in health care. More knowledge regarding this topic could help improving care with targeted measures. However, we find it too ambitious to estimate new variables and do further analyses as a part of this study and appreciate that the reviewer supports this view. However, we will certainly consider this in further studies.
Reviewer: 2 Dr. Flemming Skjøth, Aalborg University Hospital	
Comments to the Author: The authors Dr. Hetlevik et al present a study on measure for continuity of care based on Norwegian nationwide register	

data.	
2.1 The authors argue that continuity of care should(/could) be regarded as a feature of the GP-practice to which the individual patient/citizen is related, and it is argued that traditionally continuity of care is viewed as a patient level feature. From an epidemiological point both levels may be relevant, but importantly they reflect different questions. If the focus is on patient administration and service on GP level, then indeed a GP-level index seems appropriate.	We agree on this comment, and see that the two measures have different use, answering different questions. We might in our text have too much focus of which “is the best” overall, but that is not our intention and we have made some minor changes in text to reduce this impression, hopefully. See page 18 & 20 Our intended contribution is pointing out a major weakness of the UPC patient index, while at the same time defining a new COC index that incorporates GP availability.
2.2 It is not entirely clear from the text how GPs and practices are treated, the text could be strengthened by being more consistent on use of practice and GP list. Perhaps you could provide some additional information on the number of practices and fix the proportion of 1-GP practices. Which may be relevant for the evaluation of the change of not being assigned to the regular GP or practice.	The wording “GP list” was used when referring to list population and this (“..population”) is now added to clarify. Else, GP list is the short form of the index that are kept unchanged. GP practice refers to practices of the individual GP who is responsible for his/her patient list and to whom the patient data are linked to. The “group practice” is used when referring to the GP office with more GPs. We hope our minor revisions are making this clearer. In the introduction we state that 95% of GPs are working in group practices, with a reference. Based on available data we cannot link the GP to a group practice and give a detailed description on practice sizes for those who are in group practices. Based on other sources (reference 21) we generally know that half of the practices have 2-3 GP, the other half 4 or more. Although this possibility is lacking there is a fee for consultation with patients listed to a GP from outside the GP’s group practice and in a few extra cases, and this fee are that is used in 0,6% of consultation in our material. We there emphasize in this new version that Other GP in vast majority of cases another GP in the preferred GP’s practice. “siø new sentence in the results (Page 19) and discussion (page 20) The list patient system include practically the whole population, to be more precise but we reformulate also in the introduction “every inhabitant” to “more than 99% “ since there is a tiny, slightly varying proportion below 1% that do not have a regular GP. (see page 5)
2.3 The suggested index UPC^GPlist is the average number of visits at the regular GP among total number of visits excluded the visits of the current patient, as opposed to the index UPC^patient being based only on	As the reviewer writes, excluding the patients’ own data in the index is well-thought by us, as a principally correct method to disentangle the personal characteristics from the measure of care given.

the patients own visits. The principle of excluding the patients own data in the index is well-thought, although for a practice/GP with moderate amount of patients (average approx. 220) then the index would probably not vary much between patients within practice/GP. Hence a sensitivity analyses could suggested on a simpler index being just the GP level average number of visits at regular GP.	We agree that that in larger samples, removing one person's data does not affect the index. As suggested, we have performed a sensitivity test and it did not change the results. This is commented upon in the part describing statistics.
2.4 The index is calculated over a two-year period 2016-17 and assumed indicative for the service in 2018 in which the individual level outcomes are observed. This seems appropriate for this purpose.	We agree.
2.5 The comment p9 l7 must relate to both indexes.	We agree and have changed the text accordingly. See page 10.
2.6 In the description of included covariates a number of non-categorical covariates are listed, it should for transparency of the analysis be specified how these are included. As they are potential confounders their effects are of no interest in respect to the manuscript, a flexible approach could then be applied such as splines for non-categorical covariates.	In the subsection "Statistical analysis" (page 11-12), we have specified how non-categorical covariates are included. In result tables 2-4, all non-categorical covariates are in level form. In a supplementary file we report results from using linear splines models (which did not change results qualitatively).
2.7 It is not clear what the purpose of the comment regarding municipality-fixed effects is, are the included and if so what are they.	Our choice of estimator is elaborated upon in the Statistical analysis section, (page 11) with a reference to Gunasekara et al. (2014). Municipality-fixed effects are included in all estimations, as specified in notes to the tables.
2.8 Furthermore as the analyses are clustered at practice/GP level then these effects are to some extent already accounted for.	We chose to cluster at GP level additionally, to account for different personal "styles" of practice as a personal feature of the GP. Such variation might impact COC offered by the GP independently of the surroundings represented by correction for municipality, as described in pkt 2.7 above. We have added some information about this in the text in Statistics section, see page 12.
2.9 It also need detailed how the exposure index UPC actually enters the model, I presume a linear relation is assumed. Was this assumption evaluated?	For both indices, a linear relation is assumed. This assumption is in line with other COC studies, and facilitates a comparison of results. We have now clarified this point in the Statistical Analysis section.
2.10 As the UPC^GP index is a GP level index, I think the table 1 presentation could be strengthened by giving more information on GP level as eg number of practices/GP, number of patients per	Unfortunately, our data does not include information that allows us to identify practices. We agree on the reviewer's last comment here and have now removed the outcomes from table 1.

practices/GP, number of GPs per practice. Listing the outcomes in table 1 is inappropriate in my view, you also repeat them in table 2.	
2.11 As regards table 2 all listing of the effects of patients characteristics should be taken out, they are of no interest for your investigation of the effects of the continuity of care index, and in general you have no guarantee that the study design and other included covariates would ensure that the parameters have the anticipated interpretation.	As suggested, we have kept all patient characteristics as controls in the model specification, but removed them from the result table 2, as also suggested by another reviewer.
2.12 The analysis represented by results in table 3 should be described in the methods section, I assume they are obtained by stratified analyses.	Thank you for pointing this out. You are right that these are stratified analyses. We have now added this information in the method section. See page 12 and text related to table 3
2.13 A final explorative analysis is presented, it is not clear what the purpose of these are and they seem to have another purpose than these primary exposure index. Perhaps it should be clarified whether the indexes analyzed here are patient characteristics or practice/GP characteristics.	Thank you for suggesting this clarification. We have added a description on the two explanatory variables are estimated and used in these final analyses to study effect of different reasons for dis-continuity (see page 10, 12). We have also added information on the purpose/focus in the second last paragraph before table 4 at page 17 and made some changes in line with these description in the table 4. We have also included this research question among the aims of the study (last paragraph in the Introduction, page 7).
2.14 Minor What is ACSC?	This is a short form of "ambulatory care sensitive conditions". This abbreviation is introduced in the introduction, and also explained in the footnotes of the tables.
2.15 It disturbs that the tables 2-4 although being structured likewise then the rows are permuted in table 4.	Thank you for making us aware of this. We have changed table 4 to make the order of rows similar to tables 2 and 3.
Reviewer: 3 Dr. Muhammad Faisal, University of Bradford	
Comments to the Author: 3.1 Hetlevik and Holmås aimed to explore the association of continuity of care with hospital admission and mortality. Overall study has used appropriate statistical methods, but the results can be presented in better form, for example, distribution of covariates and their relationship with outcomes can be reported.	We accept that results can be presented differently but have chosen not to go more in-dept regarding covariates. We have also removed them from tables in-line with the suggestion below, that also are suggested by another reviewer
3.2 Following are my detailed comments: I thought a research question was to explore the association of two COC indices (conventional vs proposed) with outcomes	Thank you for this clear advice. The formulation of the aim was not clear. We have rephrased aims in the last paragraph of the introduction. See page 7. The motivation for

such as mortality and hospital admission. However, suddenly exposure is changed to 'Education level'. A well-defined research question is needed.	doing stratified analysis by education (and also by gender, not tabulated) is given in the introduction, 4th paragraph. We have also given more details explaining the models used for the results presented in table 3 and 4. See page 10 and 12) We do not use gender or education as an exposure, but performed stratified analyses to see if the associations between UPCGPlist and our outcomes varies between different social demographic groups. We have as in our previous version reported results from the stratified analyses for gender in the text and for education in table 3.
3.3 I appreciate this study just explore association, which is different from a causal relationship, but interpreting all model coefficient will produce biased results and it is well known as Table 2 fallacy Daniel Westreich, Sander Greenland, The Table 2 Fallacy: Presenting and Interpreting Confounder and Modifier Coefficients, American Journal of Epidemiology, Volume 177, Issue 4, 15 February 2013, Pages 292–298, https://doi.org/10.1093/aje/kws412	Thank you for this useful comment and reference. We have in our manuscript a clear focus on UPC as explanatory variable, and do not discuss other variables as such. We might have made the ground for a misunderstanding concerning the use of education as an explanatory variable (as stated above, 3.2), since our description of statistic models used were not clear. Sorry for that. Following the advice from you and another reviewer, we have kept all patient characteristics as controls in the model specification but removed them from the result table 2.
3.4 Ideally, table 1 shall split up by outcome. How the relationship of outcome with covariates has been explored? Boxplot or scatter plots	We have removed the outcomes from tab 1 and this is now a description of variables used in regression models. As suggested, we have used scatter plots to visualize the relationship between emergency admissions and four different continuous variables (age, total income, number of GP visits, and number of hospital admissions). These plots are included in a separate file to the editor/reviewers. However, in this context, these plots are not very informative, since one "dot" may represent one or a very high number of observations, and the reader cannot tell the difference. Therefore, they are not included in the paper. In the Appendix, we now report results from estimating a piecewise linear function, related to the above-mentioned covariates.

VERSION 2 – REVIEW

REVIEWER	Baker, Richard University of Leicester, Health Sciences
REVIEW RETURNED	13-Sep-2021

GENERAL COMMENTS	My concerns have been largely addressed in the revisions. I am not a statistician and note the feedback on statistical issues in the previous review. These appear to have been addressed but this needs confirmation by a statistics expert. I don't have further revisions to request and from my point of view the paper is now suitable for publication.
--

REVIEWER	Skjøth, Flemming Aalborg University Hospital
REVIEW RETURNED	10-Sep-2021

GENERAL COMMENTS	The authors have given adequate and sensible response to the queries in my review and the manuscript have improved with better focus on the addressed question. Reading the manuscript again it however strikes me that the central index'es are only described in words. Although they are simple summations, the reader now cannot find a formula for these key concepts. Thus, this is hereby suggested. Superscript (-i) is often used to represent omission of observation i. The statistical model is a linear random effects model with GP as a random effect. I agree on the inclusion of municipality to adjust for confounding factors associated with municipality, but whether it is included as a fixed or random effect is actually a matter of generalizability of the results. If included as a fixed effect, the results are representative for the included municipalities, whereas if as a random effect, the results are representative for population of municipalities. The latter interpretation is thus wider than the former. It does not come for free, the prize wider confidence intervals on your exposure estimates. As you already include GP a random effect, I think you should take the step also to include municipality as random effect, and hence the model would be considered as a multilevel linear random effects model. In the current model description in section Statistical analysis I suggest to move the discussion on municipality to the end and join with the comments on the inclusion of GP as a random effect / cluster. I still would prefer a sentence on that ordinal covariates, like age, income etc enter the model as linear effects; instead of the sentence 'The non-categorical .. variables are included in level form ...' which is a bit ambiguous. Followingly it can be noted that such linear effects were replaced with a splines representation. A final comment regarding the last sentence in the first paragraph "Therefore, we have Non-linear models without fixed effects." The last part is not really meaningful as you never would consider non-linear random effects models, and it is of course not what you mean. I suggest you remove the 'without fixed effects' part.
---

VERSION 2 – AUTHOR RESPONSE

Reviewer: 2

Dr. Flemming Skjøth, Aalborg University Hospital

Comments to the Author:

The authors have given adequate and sensible response to the queries in my review and the manuscript have improved with better focus on the addressed question.

Reading the manuscript again it however strikes me that the central index'es are only described in words. Although they are simple summations, the reader now cannot find a formula for these key concepts. Thus, this is hereby suggested. Superscript (-i) is often used to represent omission of observation i.

R: Thank you for this suggestion. We have added a formal definition of the two indices in the section "Continuity of care measures".

The statistical model is a linear random effects model with GP as a random effect. I agree on the inclusion of municipality to adjust for confounding factors associated with municipality, but whether it is included as a fixed or random effect is actually a matter of generalizability of the results. If included as a fixed effect, the results are representative for the included municipalities, whereas if as a random effect, the results are representative for population of municipalities. The latter interpretation is thus wider than the former. It does not come for free, the prize wider confidence intervals on your exposure estimates. As you already include GP a random effect, I think you should take the step also to include municipality as random effect, and hence the model would be considered as a multilevel linear random effects model.

In the current model description in section Statistical analysis I suggest to move the discussion on municipality to the end and join with the comments on the inclusion of GP as a random effect / cluster.

R: We see that our choice of estimator needs a clarification with regards to terminology. In econometrics, "fixed effect" models, relevant for panel data, rely only on variation within individuals and hence are not affected by confounding from unmeasured time-invariant factors. This definition differs from the usage in statistics, a distinction which is made explicit in one of the articles we have referred to in our previous version, Gunasekara et al. (2014). In the revised "Statistical analysis" paragraph, we have elaborated on the characteristics of the estimator used and explicitly commented on terminology.

I still would prefer a sentence on that ordinal covariates, like age, income etc enter the model as linear effects; instead of the sentence 'The non-categorical .. variables are included in level form ...' which is

a bit ambiguous. Followingly it can be noted that such linear effects were replaced with a splines representation.

R: As suggested, we have changed the wordings.

A final comment regarding the last sentence in the first paragraph “Therefore, we have Non-linear models without fixed effects.” The last part is not really meaningful as you never would consider non-linear random effects models, and it is of course not what you mean. I suggest you remove the ‘without fixed effects’ part.

R: We assume this comment is based on the difference in terminology which we refer to above.

VERSION 3 – REVIEW

REVIEWER	Skjøth, Flemming Aalborg University Hospital
REVIEW RETURNED	29-Oct-2021

GENERAL COMMENTS	I have only some comments regarding the description of the statistical model specifically the use of terminology. I do admit I have not met the econometrics terminology before, but apparently all agree it is confusing with same term used for different models. Using a dummy parameter or mean adjusting to account for within cluster time-invariant confounding is of course well known. The statistical analysis section does not read well, I think. The municipality effect modelling is not really important for central problem. I would recommend you to drop the focus on the term 'fixed effects' as it just confuse. It is then not necessary with the added long discussion on difference between econometric and statistics lingo. The reference to Gunasekara is fine to keep. As regards inclusion of municipality I think it is enough to comment that municipality is added to the model as a factor in order to account for time-invariant unmeasured counfounders on municipality level - or something like that. Likely the problems of the logit model relate to the inclusion of a factor with 4000 levels(=parameters) among some with very few or many events.
--